# Long-term Surface Temperature (LoST) Database as a complement for GCM preindustrial simulations

Francisco José Cuesta-Valero[1,2], Almudena García-García[1,2], Hugo Beltrami[2], Eduardo Zorita[3], and Fernando Jaume-Santero[2,4]

[1]Environmental Sciences Program, Memorial University of Newfoundland, St. John's, NL, Canada.
[2]Climate & Atmospheric Sciences Institute, St. Francis Xavier University, Antigonish, NS, Canada.
[3]Institute of Coastal Research, Hemlholtz-Zentrum Geesthacht, Germany.
[4]Departamento de Física de la Tierra y Astrofísica, Facultad de Ciencias Físicas, Universidad Complutense de Madrid, 28040, Madrid, Spain.

**Correspondence:** H. Beltrami, Climate & Atmospheric Sciences Institute, Department of Earth Sciences, St. Francis Xavier University, 5009 Chapel Square, Antigonish, NS, B2G 2W5, Canada (hugo@stfx.ca).

**Abstract.** Estimates of climate sensitivity from General Circulation Model (GCM) simulations still present a large spread despite the continued improvements in climate modeling since the 1970s. This variability is partially caused by the dependence of several long-term feedback mechanisms on the reference climate state. Indeed, state-of-the-art GCMs present a large spread of control climate states probably due to the lack of a suitable reference for constraining the climatology of preindustrial simulations. We assemble a new gridded database of long-term ground surface temperatures (LoST database) obtained from geothermal data over North America, and we explore its use as a potential reference for the evaluation of GCM preindustrial simulations. We compare the LoST database with observations from the CRU database, as well as with five past millennium transient climate simulations and five preindustrial control simulations from the third phase of the Paleoclimate Modelling Intercomparison Project (PMIP3) and the fifth phase of the Coupled Model Intercomparison Project (CMIP5). The database is consistent with meteorological observations as well as with both types of preindustrial simulations, which suggests that LoST temperatures can be employed as a reference to narrow down the spread of surface temperature climatologies on GCM preindustrial control and past millennium simulations.

## 1 Introduction

General Circulation Model (GCM) simulations of the Earth's climate are sophisticated tools that reproduce many physical processes of the climate system, helping to understand and characterize the dynamics of the climate system both at global and regional scales, as well as from decadal to millennial timescales (Flato et al., 2013). Despite the large number of different GCMs developed and maintained by modeling groups around the world, future projections of climate change still present a large degree of uncertainty (Knutti and Sedláček, 2012), mainly due to the different climate sensitivity achieved by each model. The Equilibrium Climate Sensitivity (ECS) is typically defined as the change in equilibrium temperature given a doubling of atmospheric $CO_2$ concentration (Gregory et al., 2002), and it is considered one of the most important metrics to understand the long-term evolution of the climate system. Numerous studies, nonetheless, have unsuccessfully tried to narrow the uncer-

tainty range of ECS using observations, simulations and paleoreconstructions, the best estimates of ECS remaining between $1.5 - 4.5\ °C$ since the 1970s (Knutti et al., 2017).

The large uncertainty in ECS estimates is also present in state-of-the-art GCMs (Andrews et al., 2012; Flato et al., 2013; Forster et al., 2013; Knutti et al., 2017), mainly as result of approximating the description of several climate phenomena, tuning practices, and the spread in control climate states. Each GCM approximates and resolves the differential equations governing the evolution of the climate system using different numerical methods and algorithms, leading to a diverse representation of the climate evolution within the array of models (Dommenget, 2016). Additionally, each GCM employs different parameterizations for approximating processes that cannot be resolved due to the lack of spatial resolution or computational resources, such as radiative transfer, convection or clouds (McFarlane, 2011; Sen Gupta et al., 2013). All these necessary approximations add inconsistencies to simulations, affecting the simulated climate state and trajectory. Parameterizations of radiative forcing by $CO_2$ in climate models are of special importance, being responsible for nearly 50% of the uncertainties in the estimated values of ECS (Soden et al., 2018). Another practice related to parameterizations that affects the simulated ECS is model tuning (Mauritsen et al., 2012; Hourdin et al., 2017; Schmidt et al., 2017). Tuning practices consist in varying model parameters, whose values are poorly constrained by theory or observations or not constrained at all, to obtain a simulated climate evolution compatible with observations. Thereby, this parameter adjustment affects the representation of feedback mechanisms and other physical processes within the model, altering the response to external forcings (Mauritsen et al., 2012; Schmidt et al., 2017).

Furthermore, the magnitude of some important long-term feedback mechanisms depends on the mean climate state - i.e., the model response to external forcings is itself mean state dependent (Dommenget, 2016; Hu et al., 2017, and references therein). Ice-albedo and water vapor feedbacks are two important processes affected by the control climate state (Hu et al., 2017). The strength of both feedbacks is associated with simulated absolute values of surface temperature, since absolute temperature is the main factor governing water phase changes on the Earth. Permafrost stability, and thus permafrost carbon feedback, also depends on the reference climatology and the simulated climate trajectory (Slater and Lawrence, 2013). Although many GCMs are still in the process of implementing permafrost modules in their code, several studies have suggested that the impact of the permafrost carbon feedback on climate evolution would be important (e.g., Koven et al., 2011; MacDougall et al., 2012). Therefore, a constrained preindustrial control simulation may improve the representation of those feedbacks in transient climate experiments, reducing the uncertainty of ECS estimates from model simulations, as well as reducing the spread in projections of future climate change (Dommenget, 2016; Hu et al., 2017). At this point, estimates of preindustrial long-term surface temperatures from geothermal data may be a useful reference for assessing wether the simulated surface temperature climatology is realistic enough in preindustrial climate simulations. Additionally, such preindustrial long-term absolute temperatures may be employed to define a preindustrial baseline from which to evaluate the temperature change due to the anthropogenic influence on climate (Hawkins et al., 2017).

Borehole Temperature Profile (BTP) measurements have been employed to estimate both global and regional past trends of surface temperature (e.g., Vasseur et al., 1983; Huang et al., 2000; Harris and Chapman, 2001; Beltrami, 2002; Beltrami and Bourlon, 2004) and surface flux histories over the last centuries (e.g., Wang and Bras, 1999; Beltrami, 2002; Beltrami et al., 2002, 2006; Demezhko and Gornostaeva, 2015a, b). Several studies have validated the borehole methodology using

past millennium simulations from the ECHO-G GCM (González-Rouco et al., 2006, 2009) and the PMIP3/CMIP5 GCMs (García-García et al., 2016), reinforcing results retrieved from subsurface temperature. Reconstructions of surface temperature and surface flux from borehole measurements have been compared with ECHO-G millennial simulations (Stevens et al., 2008; MacDougall et al., 2010), as well as with estimates of continental heat storage from CMIP5 GCM simulations (Cuesta-Valero et al., 2016). All these direct comparisons between BTP estimates and GCM simulations have revealed several strengths and weaknesses of GCM simulations, and have contributed to the improvement of the represented physical processes relevant for the climate evolution within land surface model components (e.g., Alexeev et al., 2007; MacDougall and Beltrami, 2017).

Here, we propose the use of long-term surface temperatures estimated from BTP measurements as an additional tool to better evaluate the realism of surface temperature climatology within GCM preindustrial simulations, and thereby to help to improve the representation of mean state dependent feedbacks. These long-term surface temperatures are retrieved from the quasi-equilibrium state of the subsurface thermal regime at the location of each BTP measurement. This is estimated from the deepest section of the temperature profile, which is the part least affected by the recent changes in the surface energy balance. The subsurface temperature at the bottom part of each temperature profile depends linearly on depth, and the extrapolation of this linear behavior to the surface is interpreted as the long-term mean surface temperature at each borehole site (e.g. Huang et al., 2000; Harris and Chapman, 2001; Beltrami, 2002). We present here a gridded Long-term Surface Temperature (LoST) database for most of continental North America and three Caribbean islands (Cuba, Hispaniola and Puerto Rico) using 514 BTP measurements. This database is freely available for the scientific community at https://figshare.com/s/f20d6002a57cf3279a1c. The database is compared with five past millennium and five preindustrial control simulations from the PMIP3/CMIP5 archive to assess the realism of the simulated preindustrial equilibrium state by the current generation of global climate models.

## 2 Data

### 2.1 Meteorological data: Climate Research Unit (CRU) data

We employ surface air temperatures from the University of East Anglia Climatic Research Unit's (CRU) TS4.01 gridded dataset (Harris et al., 2014) for evaluation purposes. This dataset consists in a gridded set of climate variables derived from meteorological observations worldwide. Sources of meteorological data include several national meteorological services, CRU archives, the World Meteorological Organization (WMO) and the National Oceanic and Atmospheric Administration (NOAA). Surface air temperatures are supplied on a monthly resolution for continental areas except for Antarctica from 1901 to 2016 of the Common Era (CE).

### 2.2 GCM data

We use five Past Millennium (PM) and five preindustrial control (piControl) GCM simulations (see Table 1 for references) from the third phase of the Paleoclimate Modelling Intercomparison Project and the fifth phase of the Coupled Model Intercomparison Project (PMIP3/CMIP5) (Braconnot et al., 2012; Taylor et al., 2011) to test the LoST database. PM simulations (Past1000

experiment in the PMIP3/CMIP5 database) simulate the climate response to prescribed external forcings from Schmidt et al. (2011) for the period 850-1850 CE, including orbital variations, changes in solar activity, greenhouse gas concentrations, volcanic eruptions and changes in land use and land cover. Each PMIP3/CMIP5 GCM also performs a piControl simulation forced with agreed preindustrial forcings to provide a baseline from which to start transient climate experiments. For more details about the PMIP3/CMIP5 control simulations and initialization procedures see Sen Gupta et al. (2013) and Séférian et al. (2016).

## 2.3  Borehole data

Here, we use estimates of long-term surface temperatures from the database described in Jaume-Santero et al. (2016). The BTP measurements of this database have been previously filtered excluding profiles with non-climatic signals and artifacts, thus providing 514 BTPs suitable for climate studies over North America. The standard methodology to retrieve past temperature and flux histories from geothermal data assumes that each borehole temperature profile results from the superposition of a transient perturbation due to the changes in the surface energy balance with time and the quasi steady-state of the subsurface thermal regime (e.g. Huang et al., 2000; Harris and Chapman, 2001; Beltrami, 2002). Therefore, considering the subsurface as a half space without heat production from radioactive decay or advection, the solution of the heat diffusion equation for a temperature profile can be approximated as (e.g., Jaume-Santero et al., 2016):

$$T(z) = T_0 + q_0 \cdot R(z) + T_t(t), \tag{1}$$

where $T_t$ is the surface transient perturbation, $T_0$ is the long-term surface temperature ($T_0$ temperature hereafter), $q_0$ is the subsurface flux at equilibrium and $R(z)$ is the thermal resistance (Bullard and Schonland, 1939). Estimates of thermal resistance require measurements of thermal conductivity through the subsurface profile, but the majority of available BTPs does not present such conductivity data. Thus, the thermal conductivity is assumed to be constant and Eq. (1) is rewritten as

$$T(z) = T_0 + \Gamma \cdot z + T_t(t), \tag{2}$$

where $\Gamma$ is the subsurface thermal gradient at equilibrium. The recorded surface transient perturbation ($T_t$) can be retrieved from each temperature profile, once the subsurface thermal equilibrium is estimated (for more details about the borehole methodology, see Mareschal and Beltrami 1992; Bodri and Cermak 2007; Jaume-Santero et al. 2016). As the heat flux from the Earth's interior remains stable at time scales of millions of years and the deepest part of a BTP is the least affected by the recent changes in the surface energy balance, the quasi-equilibrium state of the subsurface thermal regime can be estimated from the deepest temperatures of each borehole profile (see scheme in Fig. 1). Once vertical variations in thermal properties of the subsurface rocks are taken into account, temperature depends linearly on depth at the bottom part of the temperature profile, allowing to approximate the subsurface thermal equilibrium by a linear least-squares regression. The extrapolation of this linear behavior to the surface can be interpreted as the long-term mean surface temperature at each borehole location ($T_0$ temperature in Eq. 2 and Fig. 1, see Pickler et al. 2016 for further details). Depending on the profile's depth, the $T_0$ temperatures represent the long-term ground surface temperature for a determined period of time. Due to the nature of heat diffusion through the

ground, the required time (t) for a change in the surface energy balance to reach a certain depth (z) is given by (Carslaw and Jaeger, 1959; Pickler et al., 2016):

$$t \approx \frac{z^2}{4\kappa},\tag{3}$$

where $\kappa$ is the thermal diffusivity of the subsurface. We consider $\kappa = 1 \times 10^{-6}$ m$^2$s$^{-1}$ for all BTP measurements (Cermak and Rybach, 1982). In this study, all BTPs are truncated at the same depth (300 m) to ensure that all $T_0$ temperatures are estimated for the same temporal period. We use the last hundred meters of each BTP to estimate the subsurface thermal equilibrium, obtaining an estimated temporal period that approximately ranges from $\sim 1300$ CE (z = 300 m) to $\sim 1700$ CE (z = 200 m). Thereby, this period of time provides a baseline to compare with long-term temperatures from the PMIP3/CMIP5 PM simulations. However, the estimated temporal period is not homogeneous as result of the non-linear relationship between time and depth (Beltrami and Mareschal, 1995), and thus estimates of recent years (i.e., 1700 CE) are better determined than estimates of past years (1300 CE). Influences of long-term perturbations of the past surface energy budget outside of that temporal window may also affect the temperature within the depth range used here, see Section 5 for more details.

## 3 The LoST database

In order to provide with a gridded dataset over continental North America, $T_0$ temperatures from BTP measurements are spatially interpolated to a $0.5° \times 0.5°$ grid using the Gradient plus Inverse Distance Squared (GIDS) technique. The GIDS method (Nalder and Wein, 1998) relies on the multiple linear regression of observed climate variables to retrieve longitudinal, latitudinal and altitudinal gradients that are employed to estimate values for gridded nodes. The contribution of each measurement is inverse-weighted by their squared distance to the target node, while the coefficients from the regression analysis allow to correct for the location of each measurement:

$$V_0 = \frac{\sum_{i=1}^{N} \left[ V_i + (lat_0 - lat_i)C_{lat} + (lon_0 - lon_i)C_{lon} + (z_0 - z_i)C_z \right] d_i^{-2}}{\sum_{i=1}^{N} d_i^{-2}},\tag{4}$$

where $V_0$ is the predicted variable at the target node, $V_i$, $lat_i$, $lon_i$ and $z_i$ represent the variable, latitude, longitude and altitude of the $i^{th}$ measurement respectively, $lat_0$, $lon_0$ and $z_0$ represent the latitude, longitude and altitude of the target node respectively, $C_{lat}$, $C_{lon}$ and $C_z$ are the coefficients from the regression analysis, and $d_i$ is the distance from the $i^{th}$ measurement to the target node. The propagation of known errors in the GIDS algorithm is described in Section S1. The GIDS technique has been used to interpolate surface temperature, precipitation, evapotranspiration and other climate variables in several zones of the world including North America (e.g., Price et al., 2000; Mardikis et al., 2005). Furthermore, the GIDS method performs well in comparison with other broadly used interpolation techniques like co-kriging or smoothing splines (ANUSPLIN suite) (Nalder and Wein, 1998; Price et al., 2000; Li and Heap, 2011), and it has been previously employed to downscale CMIP5 simulations (McCullough et al., 2016).

Since the $T_0$ dataset employed here provides latitudes and longitudes for each temperature profile, we expand the database estimating the altitude above sea level for each BTP measurement from the second version of the 2-minute Gridded Global

Relief Data (ETOPO2) of the National Oceanic and Atmospheric Administration (National Geophysical Data Center, 2006. Two-minute Gridded Global Relief Data (ETOPO2) v2. National Geophysical Data Center, NOAA. doi:10.7289/V5J1012Q [last accessed on July 7th, 2017]). For this study, the regression analysis of $T_0$ temperatures considering latitude, longitude and altitude yields robust results, with a $R^2$ value of $0.865$ and a p-Value $\ll 0.05$. The distance from the measurements to the nodes is computed using the Vincenty's formula for an ellipsoid with different major and minor axes (Vincenty, 1975), and therefore the altitude of both measurements and grid nodes are not considered in our distance calculations.

We performed a pseudo-proxy experiment (e.g., Smerdon, 2012) to determine which is the maximum appropriate distance from a grid node to a BTP measurement to interpolate the $T_0$ temperatures. That is, we use the long-term mean ground surface temperatures for the period 1300-1700 CE from the five PMIP3/CMIP5 PM simulations as surrogate realities, and apply the interpolation methodology employed to create the LoST database. Thereby, these GCM simulations were regridded to a $0.5° \times 0.5°$ grid, considering grid cells containing BTP measurements as reference for applying Eq. 4 to the rest of grid cells. Then, Root-Mean Squared Errors (RMSEs) between the interpolated data and the remapped simulations were computed (Fig. S1). We set $650\,\mathrm{km}$ as maximum distance criterion since this is the maximum distance at which the RMSE is lower than $1.0\,°\mathrm{C}$ for the five simulations. Such distance criterion, nevertheless, produces results for three grid cells in the Yucatan peninsula (Mexico), which we consider unjustifiable as there are no BTP measurements in or near that part of Mexico. Those grid cells are therefore masked out from our analysis.

## 4  Results

The distribution of LoST temperatures at grid cells containing BTP measurements reproduces the shape of the distribution of raw $T_0$ temperatures (Fig. 2a), indicating that the GIDS interpolation does not substantially modify the shape of the original distribution of temperatures retrieved from BTP measurements. However, the distribution of the entire LoST database resembles the distribution of CRU temperatures, differing from the distribution of the raw $T_0$ temperatures. This change in the temperature distribution after the spatial interpolation may be related to the inclusion of interpolated temperatures at higher and lower latitudes than the raw $T_0$ temperatures, as the majority of BTP measurements cover from $35\,°\mathrm{N}$ to $60\,°\mathrm{N}$. Nonetheless, the latitudinal mean temperatures from the LoST database are consistent with $T_0$ temperatures from BTP measurements, either considering only grid cells with BTP measurements or the entire LoST database (Fig. 2b). The latitudinal mean temperatures from the LoST database reach higher values than the CRU database at latitudes higher than $\sim 50\,°\mathrm{N}$, while both datasets achieve similar mean temperatures at lower latitudes (Fig. 2b). Previous studies have found warmer ground temperatures than air temperatures in meteorological observations over North America, probably due to the insulating effect of snow cover during winter (e.g., Beltrami and Kellman, 2003; Smerdon et al., 2003). That is, warmer temperatures should be expected for the LoST database than for the CRU database, as our results show (Fig. 2a and 2b). It should be noted, nevertheless, that the CRU database covers a period with a marked global temperature increase (Hartmann et al., 2013). Therefore, estimates of long-term surface temperatures from CRU data reflect such temperature increase, hindering the direct comparison between both datasets. Despite this difference in the climatology of both databases, the long-term surface temperature from the LoST dataset repro-

duces the expected spatial pattern of temperatures for North America (Figs. 2c and 2d), in agreement with long-term surface temperatures estimated from BTP measurements and with long-term surface temperatures from CRU data.

The LoST temperatures were also compared with long-term surface temperature estimates from five Past Millennium (PM) and five piControl simulations (Table 1) included in the PMIP3/CMIP5 archive to test the realism of forced and control GCM simulations in reproducing estimates of long-term surface temperatures. Long-term surface temperatures from the PM simulations are estimated as the mean surface air temperature for the period 1300-1700 CE ($SAT_0$) and the mean ground surface temperature linearly interpolated at $1.0$ m depth for the same period ($GST_0$), in order to be consistent with the estimated temporal range for $T_0$ temperatures in Section 2.3. The PMIP3/CMIP5 simulations are interpolated onto the grid of the LoST

database; $SAT_0$ and $GST_0$ values are estimated only at grid cells containing LoST temperatures. $SAT_0$ and $GST_0$ values are also estimated for piControl simulations following the same method, but averaging over each entire control simulation.

Surface temperatures from PMIP3/CMIP5 PM and piControl simulations show similar latitudinal patterns to that from the LoST database, with lower temperatures at northern latitudes and higher temperatures at southern latitudes (Figs. S2 and S3).

$SAT_0$ estimates from the CCSM4, the MRI-CGCM3 and the BCC-CSM1.1 models show generally lower values than LoST temperatures for both piControl and PM simulations, while $GST_0$ estimates show higher values than LoST temperatures at high latitudes for the same GCM simulations (Figs. S4 and S5). Such result is in agreement with previous analyses of air and ground temperature relationship within GCM simulations (González-Rouco et al., 2003; González-Rouco et al., 2006; Stieglitz and Smerdon, 2007; Koven et al., 2013; García-García et al., 2016) and meteorological observations over North America

(e.g., Smerdon et al., 2003; Beltrami and Kellman, 2003). In contrast, MPI-ESM-P and GISS-E2-R simulations present lower $SAT_0$ and $GST_0$ values than LoST temperatures, indicating lower long-term ground surface temperatures than the rest of the models (Table 1 and Figs. S4 and S5). The comparison of the mean LoST temperature over North America with the simulated temperature evolution by each GCM shows three different behaviors within the PMIP3/CMIP5 ensemble. The CCSM4 and the BCC-CSM1.1 simulations present lower mean air temperatures and higher mean ground temperatures than the mean LoST

temperature (Fig. 3 and Table 1). The similar $GST_0$ and mean ground surface temperatures for the CCSM4 and the BCC-CSM1.1 GCMs in both PM and piControl simulations were expected since these models use a similar land surface model component (Wu et al., 2014) and the simulated ground temperatures by CMIP5 models are highly dependent on the employed land surface model component (Slater and Lawrence, 2013, García-García et al., submitted to *Journal of Geophysical Research - Atmospheres*). In contrast, the GISS-E2-R and the MPI-ESM-P models produce lower mean $GST_0$ values than the mean LoST

temperature and the rest of models, while simulating similar $SAT_0$ values to those from the rest of the PMIP3/CMIP5 GCMs. Previous results have shown that the MPI-ESM-P PM simulation yields a high air-ground temperature coupling (García-García et al., 2016), probably due to the omission of latent heat of fusion in soil water (Koven et al., 2013). This could cause the low ground surface temperature simulated by the MPI-ESM-P model in both PM and piControl simulations in comparison with the mean LoST temperature (Fig 3). A strong air-ground coupling may also cause the low ground surface temperature in the GISS-E2-R simulations, since the magnitude of the difference between $GST_0$ and $SAT_0$ is similar to that from the MPI-ESM-P simulations (Table 1). Finally, the MRI-CGCM3 PM simulation yields $GST_0$ values below the LoST climatology, but only by $0.3\ °C$ ($0.1\ °C$ if considering the $2\sigma$ range of the LoST climatology, Fig. S6), which are relatively small in comparison with

5   the differences between the LoST climatology and the $GST_0$ values from MPI-ESM-P and GISS-E2-R simulations ($> 2.0\,°C$, Table 1). Thus, we can consider that three of the five PMIP3/CMIP5 GCMs (the CCSM4, the MRI-CGCM3 and the BCC-CSM1.1) simulate a surface temperature climatology, in the PM (1300-1700 CE) and piControl simulations, comparable to that from the LoST dataset, which is an unexpected result as none of the PMIP3/CMIP5 GCM simulations studied here were specifically tuned to match this climatology.

## 5   Discussion

Our results demonstrate that LoST temperatures can be used as reference for assessing the represented climatology in both PM and piControl simulations. The determination of $T_0$ temperatures, nevertheless, presents some uncertainties that should be discussed. The extrapolation of each quasi-equilibrium temperature profile to the surface introduces a small error in the LoST estimates, averaging less than $0.15\,°C$ from the 514 BTPs evaluated here (see Section S1 for details about the error treatment in the LoST database). Rock heterogeneity should also be considered for estimating $T_0$ temperatures. We assume, nevertheless, homogenous thermal properties for all borehole profiles, which is another source of uncertainty for LoST temperatures. The ideal approach consists in estimating the thermal resistance with depth (Eq. 1), but the absence of thermal conductivity measurements for the employed BTPs (Jaume-Santero et al., 2016) makes that approach impractical. Additionally, measurements of thermal conductivity tend to be distributed around a central value (e.g., the measurements at the Neil Well, Beltrami and Taylor, 1995). If the thermal conductivity varies systematically with depth at a certain location, such variation will be reflected in the temperature profile as an unphysical signal. Such logs were removed from the database employed in this analysis, as explained in Jaume-Santero et al. (2016). Therefore, it is reasonable to assume a homogenous conductivity with depth. Long-term alterations of the surface energy balance out of the 1300-1700 CE period may also affect the LoST estimates. Particularly, possible transient temperatures in BTPs due to the Little Ice Age (LIA) and the Medieval Warm Period (MWP) add a certain degree of uncertainty in the determination of $T_0$ values. However, the spatial extent of both LIA and MWP was not homogeneous over North America (e.g., Masson-Delmotte et al., 2013, and references therein), meaning that not all BTPs were affected by the events (Beltrami and Mareschal, 1992; Chouinard et al., 2007; Jaume-Santero et al., 2016). Additionally, the influence of the LIA and the MWP should be part of any millennial-scale transient climate simulation, and therefore the effect of such climate events is taken into account in the comparison between LoST results and transient climate simulations. The absence of these two periods in piControl simulations is probably contributing to the slightly poorer agreement between LoST temperatures and piControl temperatures in comparison with results for the PM simulation (Figure 3). Another factor that may impact the retrieved quasi-equilibrium temperature profile is the heterogeneity of North American topography (e.g., Kohl, 1999). To our knowledge, all analyzed BTPs are located in plain terrain, and were not corrected for elevation since the employed BTP database does not provide elevation data. Therefore, we use the ETOPO2 database to assess if the altitude distribution of BTPs is enough for representing the topography of the LoST domain. The altitude distribution over the LoST domain and at grid cells containing boreholes sites are displayed in Fig. S8. Both histograms present a similar shape for altitudes up to $\sim 430$ m, showing a lack of borehole locations at altitudes between $\sim 430$ m and $\sim 1013$ m. The uneven latitudinal distribution of borehole

sites is probably causing this gap in the distribution of altitudes, as well as a small excess of BTP locations at high altitudes. Despite these differences, both distributions are generally in agreement, indicating a sufficient altitude distribution from the
borehole database to represent the North American broad-scale topography.

There are, however, two main limitations for the application of the LoST database at this stage of the study: the supplied variable and the regional character of the database. The LoST database is constituted by estimates of long-term ground surface temperatures, while GCM simulations are typically evaluated against observations of surface air temperature (SAT) (e.g., Mauritsen et al., 2012; Flato et al., 2013; Séférian et al., 2016; Schmidt et al., 2017). We can provide a reference for simulated
long-term SAT by accounting for the offset between simulated air and ground temperatures and using the LoST temperatures. As an example, SAT references are estimated for the five PM and five piControl simulations employed in this study (dashed blue line in Fig. 3). SAT references for PM simulations are estimated from the offset between air and ground temperatures in piControl simulations, while SAT references for piControl simulations are estimated from the offset between air and ground temperatures in PM simulations. Such offsets show a constant behavior in both simulations (Fig. S7). GCM simulations in
disagreement with the estimated SAT reference (the MPI-ESM-P and the GISS-E2-R simulations) may be representing a strong air-ground coupling, as discussed in Section 4. Therefore, although the LoST database contains estimates of ground surface temperatures, it may be also used to assess simulated long-term surface air temperatures on a first order approach.

The regional character of the presented LoST database poses some caveats for analyzing the global climatology of preindustrial simulations. Indeed, results of the simulated regional climatology cannot be globally extrapolated since the magnitude
of the potential spurious drifts in control simulations varies markedly at regional scales and these regional drifts could be larger than the global-averaged drift (Sen Gupta et al., 2012, 2013). Further work would consist in generating a global LoST database from the existing global network of BTP measurements, helping to minimize the effect of possible regional drifts on the simulated climatology. However, BTP measurements are scarce in the Southern Hemisphere, a potential burden that needs to be considered for assembling such global version of this database. Additionally, the temperature profiles employed in this
study to estimate $T_0$ temperatures were truncated to $300$ m of depth, which determines the temporal period of reference for the comparison with PM simulations. Deeper BTP measurements can retrieve the climatology of previous time periods, although the global BTP network contains fewer temperature profiles deeper than $300$ m (see Fig. 1 in Beltrami et al., 2015).

Despite the regional character of the LoST temperatures, the northern BTPs contained in this database allow to evaluate the long-term stability of permafrost over North America. That is, the northern temperatures in this database can be compared
with regional and global simulations as a reference to the preindustrial permafrost stability. Furthermore, previous studies have found that the CMIP5 GCM simulations have difficulties to properly represent permafrost evolution (Koven et al., 2013; Slater and Lawrence, 2013), partially due to the broad range of simulated climate trajectories by each GCM and the differences between the employed land surface model components (Slater and Lawrence, 2013). Using LoST temperatures to improve the surface temperature climatology of global and regional simulations may enhance the simulated long-term preindustrial $0$ °C isotherm, which is important to correctly represent permafrost evolution.

Numerous proxy-data based reconstructions of temperature, precipitation and other climate related variables exist for North America, providing a reference for the evaluation of important aspects of past and future climate model simulations (e.g.,

PAGES 2k-PMIP3 Group, 2015; Cook et al., 2015). Proxy-data temperature reconstructions have already been compared against borehole temperature records of past variations in surface temperature over North America (e.g., Jaume-Santero et al., 2016). It is worthy to note that proxy systems are indirect sources of climate information requiring a calibration procedure with modern meteorological data, while borehole temperature data consist of direct measurements of the thermal regime of the subsurface in the recent past. That is, the LoST database contains information derived from direct measurements of subsurface

temperatures, constituting the first estimates of long-term absolute surface temperatures in North America. Another important difference between proxy and borehole reconstructions is that most proxy systems generally capture high-frequency variations of climate conditions (Moberg et al., 2005), while borehole temperature profiles record long-term changes in the surface conditions, filtering out short-period signals. In this context, LoST temperatures provide a complementary reference to the multiproxy database over North America for evaluating the performance of climate model simulations.

**6 Conclusions**

A gridded database of past long-term surface temperatures over most part of continental North America has been assembled from geothermal measurements. Our results show that this database can be used as reference to evaluate the realism of GCM preindustrial control and past millennium simulations and possibly to improve the reference climate state by adjusting key parameters or preindustrial forcings in control experiments. Thereby, spread in ECS estimates by GCM simulations may be

reduced given the relationship between control temperature climatology and three long-term powerful feedbacks as the ice-albedo feedback, the water vapor feedback and the permafrost carbon feedback. Future work would consist in generating a global version of the LoST database using the rest of the global network of borehole temperature profile measurements and following the described methodology, as well as generating new versions of this global database including future temperature profile measurements.

*Data availability.*  The LoST database can be downloaded from https://figshare.com/s/f20d6002a57cf3279a1c, with doi:10.6084/m9.figshare.8124887

*Author contributions.*  FJCV, AGG and HB designed the study. FJCV generated the results and pictures. FJS provided the T0 data. All authors analyzed the results, discussed the outcome, and drafted the paper.

*Competing interests.*  The authors declare no competing interests.

*Acknowledgements.* We acknowledge the World Climate Research Programme's Working Group on Coupled Modelling, which is responsible for CMIP, and we thank the climate modeling groups responsibles for the model simulations used herein (listed in Table 1 of this paper) for producing and making available their model output. For CMIP the U.S. Department of Energy's Program for Climate Model Diagnosis and Intercomparison provides coordinating support and led development of software infrastructure in partnership with the Global Organization for Earth System Science Portals. We are grateful to Dmitry Demezhko, an anonymous reviewer and Irina Rogozhina for their constructive comments. We also acknowledge fruitful discussions with J. Fidel González-Rouco on a previous version of the manuscript and we thank Timothy Osborn for his guidance on using the CRU database. This work was supported by grants from the Natural Sciences and Engineering Research Council of Canada Discovery Grant (NSERC DG 140576948), the Canada Research Chairs Program (CRC 230687), and the Canada Foundation for Innovation (CFI) to H. Beltrami. H. Beltrami holds Canada Research Chair in Climate Dynamics. A.G.G. and F.J.C.V. are funded by H. Beltrami's Canada Research Chair program, the School of Graduate Students at Memorial University of Newfoundland and the Research Office at St. Francis Xavier University.

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

**Table 1.** Model name, $SAT_0$ estimates, $GST_0$ estimates, $SAT_0$ and $GST_0$ differences with the mean LoST temperatures and references for each PMIP3/CMIP5 GCM simulation. All results in $^\circ$C. Ground temperatures for MRI-CGCM3 piControl simulation could not be retrieved from the PMIP3/CMIP5 data servers. Temperature average of the LoST database is 5.2 $^\circ$C, with a 95% confidence interval between 5.0 and 5.4 $^\circ$C.

| Model | Past Millennium | | | | | Preindustrial Control | | | | |
|---|---|---|---|---|---|---|---|---|---|---|
| | $SAT_0$ | $GST_0$ | $SAT_0$-LoST | $GST_0$-LoST | Reference | $SAT_0$ | $GST_0$ | $SAT_0$-LoST | $GST_0$-LoST | Reference |
| CCSM4 | 1.53 | 5.60 | -3.65 | 0.37 | Landrum et al. 2013 | 2.12 | 6.03 | -3.07 | 0.80 | Gent et al. 2011 |
| MRI-CGCM3 | 1.38 | 4.84 | -3.81 | -0.31 | Yukimoto et al. 2012 | 1.39 | - | -3.80 | - | Yukimoto et al. 2012 |
| MPI-ESM-P | 1.63 | 2.75 | -3.56 | -2.91 | Jungclaus et al. 2014 | 2.00 | 3.10 | -3.19 | -2.56 | Jungclaus et al. 2013 |
| GISS-E2-R | 1.96 | 3.10 | -3.23 | -2.42 | Schmidt et al. 2014 | 2.02 | 3.14 | -3.16 | -2.35 | Miller et al. 2014 |
| BCC-CSM1.1 | 0.75 | 5.39 | -4.44 | 0.22 | Xiao-Ge et al. 2013 | 1.03 | 5.58 | -4.15 | 0.42 | Wu et al. 2013 |

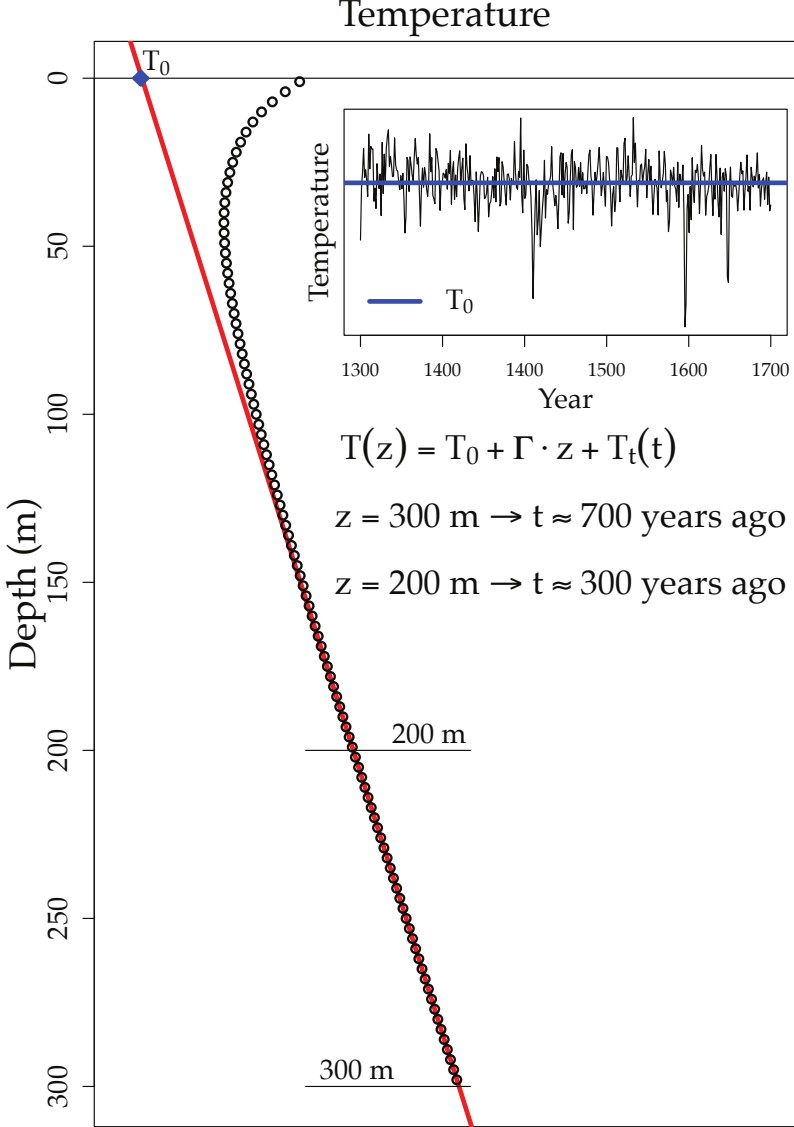

**Figure 1.** Synthetic borehole temperature profile (black dots) using data from the CCSM4 PM simulation (inset) and linear fit of temperatures between 200 m and 300 m (red line). The synthetic temperature profile is generated using the simulated global ground temperature anomaly at 1.0 m depth for the period 1300-1700 CE as transient perturbation ($T_t$), mean ground temperature as long-term surface temperature ($T_0$) and a typical thermal gradient ($\Gamma$) of $0.01\ \mathrm{K\,m^{-1}}$ (Jaume-Santero et al., 2016). The equivalence between depth ($z$) and time ($t$) is given by Eq. 3. Thermal diffusivity is considered as $\kappa = 1 \cdot 10^{-6}\ \mathrm{m^2\,s^{-1}}$ (Cermak and Rybach, 1982).

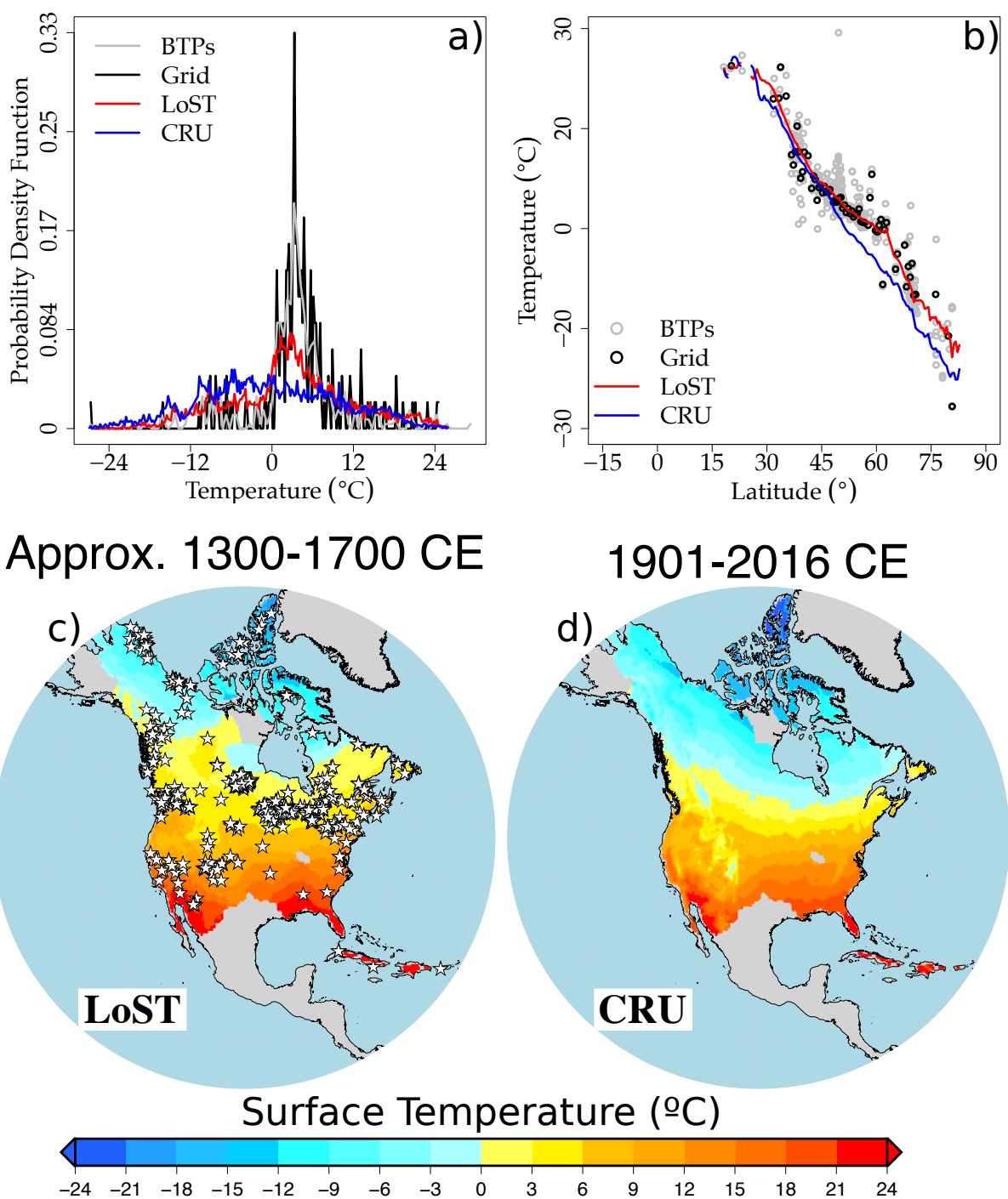

**Figure 2.** Histogram (a) and latitudinal mean temperatures (b) from BTP measurements (gray), LoST temperatures at grid cells containing BTP measurements (black), LoST temperatures (red) and mean surface air temperature from the CRU database (blue). LoST temperatures (∼1300-1700 CE) (c) in comparison with mean surface air temperature from CRU data (1901-2015 CE) (d). White stars in (c) indicate the location of the 514 BTP measurements.

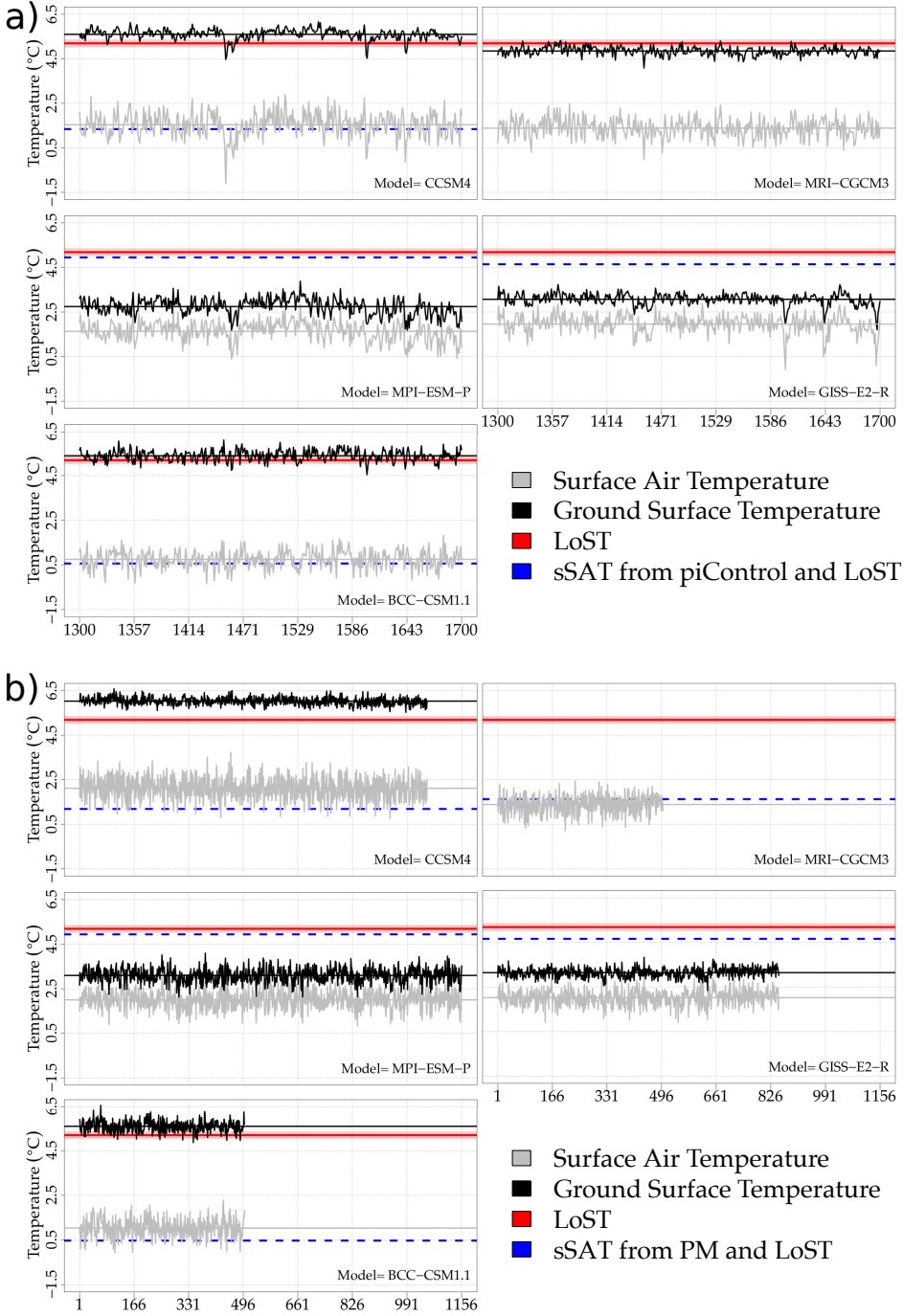

**Figure 3.** Surface air temperature evolution (gray solid line), ground surface temperature evolution (black solid line), $SAT_0$ (gray horizontal line) and $GST_0$ (black horizontal line) for (a) PMIP3/CMIP5 PM and (b) PMIP3/CMIP5 piControl simulations. Solid red lines represent the mean LoST temperature and the red shadow represents the 95% confidence interval (Section S1, Fig. S6). Dashed blue lines represent estimated references for long-term surface air temperatures from the LoST climatology and the simulated air-ground temperature offset in (a) piControl and (b) PM simulations. Ground temperatures for the MRI-CGCM3 piControl simulation could not be retrieved from the PMIP3/CMIP5 data servers.