# Peer review of "Long-term Surface Temperature (LoST) Database as a complement for GCM preindustrial simulations"

_Climate of the Past, 2018_

## Referee Comment (RC1) · Anonymous Referee #1 · 4 Jan 2019

Review of "Long-term Surface Temperature (LoST) Database as a complement for GCM preindustrial simulations" By: Cuesta-Valero, et al. Corresponding author: Beltrami

This manuscript addresses an important problem – large variation, and therefore uncertainty, in climate sensitivity estimates, that is the change in surface temperature accompanying a doubling of the atmospheric concentration of $CO_2$. Since the 1970s the variation in estimates of ECS has not improved much; still from 1.5 to 4.5 oC.

The paper introduction provides a useful summary of why the existing GCMs have not provided convergence to a narrow band of ECS: differences in model parameter-

ization, in particular radiative transfer and model tuning; feedback mechanisms such as ice-albedo and water vapor effects; and permafrost stability and permafrost carbon feedback. The authors suggest "a constrained preindustrial control simulation may improve the representation of those feedbacks in transient climate experiments, reducing the uncertainty of ECS estimates from model simulations, as well as reducing the spread in projections of future climate change."

The proposed constraint for such a preindustrial simulation is a database they have assembled called LoST (LoST = Long Term Surface Temperature). The paleo surface temperature at a given site is estimated by extrapolating a subsurface temperature profile from a depth range of 200 to 300 m, most sensitive to surface temperatures between about 1300 and 1700 CE, to the surface. The database is based on 514 temperature-depth profiles in North America. Creation of a LoST database offers the possibility of a better estimate of a preindustrial reference temperature field and thus an improvement in the ECS estimate.

The database is compared with five past millennium and five preindustrial control simulations from the PMIP3/CMIP5 archive to assess the realism of the simulated preindustrial equilibrium state by the current generation of global climate models. The paper is consistent with many previous papers in advocating that borehole temperatures are a robust complement to observational (met data) and model studies of past climate.

I recommend publishing the paper after some minor to modest revisions.

Details.

1. Appropriate databases are used. With respect to borehole temperatures, it is safe to neglect heat production (does not introduce appreciable curvature into the temperature-depth profile at the depths considered) but rock heterogeneity can be more of a problem. Extrapolation of a temperature gradient from 200 m to surface to arrive at To has an error that should be discussed.

2. Temperatures in the depth range 200 to 300 m are largely affected by surface temperatures from 300 to 700 years prior to the temperature logging as the manuscript points out, corresponding to surface temperatures from about 1300 to 1700 CE. I would like to see a comment on how much of the signal in that depth range comes from surface temperatures outside of that time window.

3. Figure 1 is a good illustration of the extrapolation of the borehole temperature profile to the surface. The term in the caption "linear fit of the last 100 m" is ambiguous. Say, "linear fit of bottom 100 m" or better still, "linear fit of temperatures between 200 and 300 m." The paper should also say that any thermal conductivity heterogeneity in the depth range 0 to 200 m would affect the zero depth (i.e. Surface) extrapolated temperature and ideally give a bound for how big an error that would introduce.

4. Figure 2. Is the temperature scale on Fig 2(b) mislabeled or is it some kind of a non-linear scale? The colored temperature scale for Fig 2(c) and (d) needs a label and units.

5. The paper would be improved by a discussion of various kinds of uncertainties in LoST and whether the magnitude of those uncertainties detract significantly from the goal of providing a robust preindustrial surface temperature field. Include: (a) extrapolation uncertainties for a typical borehole site. (b) whether the 514 sites are generally representative of the topography (elevation and site azimuth) of the region being modeled (scatter in extrapolated borehole temperatures in a region can vary by $\sim$ 4 oC).

(c) Are the BTT's in Fig 2(b) corrected for elevation or are elevation differences at particular latitude (considerable in North America) the cause of about 10 oC scatter at constant elevation?

Overall this is a refreshing new approach of showing how borehole temperature profiles can be used to complement the more conventional meteorological and GCM modeling studies to reveal the long-term evolution of surface temperature on the planet.

---

## Referee Comment (RC2) · Dmitry Demezhko (Referee) · 11 Feb 2019

Review of "Long-term Surface Temperature (LoST) Database as a complement for GCM preindustrial simulations" By: Cuesta-Valero, et al.

General comments The development and improvement of climate models (GSM) is a leading scientific method for understanding the Earth's climate system and its forecasting. Despite considerable efforts in the development of these models, there remains a large uncertainty in GCM scenarios. Authors suggest a new Long-term Surface Temperature (LoST) Database as "a reference to narrow down the spread of surface temperature climatologies on GCM preindustrial control and past millennium simula-

tions". Preindustrial (1300-1700) ground surface temperatures cover North America and obtained from borehole temperature profiles (BTP) analysis. A robust paleotemperature evaluation technique based on extrapolating the temperature profile from the interval 200-300 m to the Earth surface has been used. Unlike BTP inversion methods, this simple technique is not burdened by the uncertainties associated with the choice of the algorithm for the inverse problem solving, and provides comparable estimates of paleotemperatures. I suppose the paper describes a new and important result for the development of climatology and can be published in the CP.

Specific comments 1. The simplicity of the technique used does not obviate the need to justify it. Strictly speaking, the extrapolation of temperature profiles from the interval of 200-300 m provides a very approximate estimate of the mean ground surface temperature in 1300 - 1700. It is necessary to provide a justification or refer to the paper where it was done (for example, "First-order estimate of the GST history" technique by Pickler et al., 2016). First-order estimate technique is based on the use of formula (2), but its description"...the recorded temperature at a depth z can be related to an estimate of time (t)" is incorrect. Correctly: t is the time after which the temperature anomaly dT appeared at the surface reach 0.16dT at a depth of z and 0.005 at a depth of 2z. Therefore, if we assume that 0.16 dT is a negligible part of the anomaly, we should replace the description on Figure 1

from: $z = 300$ m -> $t \approx 1300$, $z = 200$ m -> $t \approx 1700$

to: $z = 300$ m -> $t > 700$ years ago, $z = 200$ m -> $t > 300$ years ago

2. A large sample of BTP data was used. Obviously, many temperature profiles revealed evidences of non-climatic influences within the studied interval (hydrogeology, heterogeneity of thermal properties). Did the authors select (or correct) the initial data and by what criteria?

3. P2,L32-34: "...BTP measurements have been employed to estimate ... surface flux histories over the last centuries (e.g., Beltrami, 2002; Beltrami et al., 2002, 2006)".

Here the authors refer only to themselves. Meanwhile, the possibility of estimating the surface heat flux changes from ground surface temperature changes was formulated by Wang, and Bras (1999). With regard to borehole temperature data, this technique (besides the mentioned papers) was developed in (Huang, 2006; Demezhko and Gornostaeva.2015a,b). In the last two papers an alternative measure of the Earth's climatic sensitivity has been proposed as the ratio between the ground surface flux changes and external fluxes changes. I believe that estimates of preindustrial surface heat flux changes can also be useful for GCM simulations, as well as estimates of paleotemperatures. I would like the authors to raise this question in the "Discussion" section.

References Pickler, C., Beltrami, H., and Mareschal, J.-C.: Laurentide Ice Sheet basal temperatures during the last glacial cycle as inferred from borehole data, Climate of the Past, 12, 115–127, 2016.

Wang, J. and Bras, R. L.: Ground heat flux estimated from surface soil temperature, J. Hydrol., 216, 214–226, 1999.

Huang, S.: 1851–2004 annual heat budget of the continental landmasses, Geophys. Res. Lett., 33, L04707, doi:10.1029/2005GL025300, 2006

Demezhko, D. Y., & Gornostaeva, A. A. Late Pleistocene–Holocene ground surface heat flux changes reconstructed from borehole temperature data (the Urals, Russia). Climate of the Past, 11(4), 647-652, 2015a.

Demezhko, D. Y., & Gornostaeva, A. A. Reconstructions of ground surface heat flux variations in the urals from geothermal and meteorological data. Izvestiya, Atmospheric and Oceanic Physics, 51(7), 723-736, 2015b.

---

## Author Comment (AC2) · 13 Mar 2019

References

Beltrami, H. and Taylor, A. E.: Records of climatic change in the Canadian Arctic: towards calibrating oxygen isotope data with geothermal data, Global and Planetary Change, 11, 127-138, 1995.

Bullard, E. C. and Schonland, B. F. J.: Heat flow in South Africa, Proceedings of the Royal Society of London. Series A. Mathematical and Physical Sciences, 173, 474–502, 1939.

[Figure]

Jaume-Santero, F., Pickler, C., Beltrami, H., and Mareschal, J.-C.: North American regional climate reconstruction from ground surface temperature histories, Climate of the Past, 12, 2181-2194, 2016.

---

## Author Comment (AC3) · 13 Mar 2019

**We thank Dmitry Demezhko for his thoughtful and constructive feedback.**

**Reviewer comments are shown in plain text, while author responses are shown in bold text.**

Dmitry Demezhko (Referee)

Review of "Long-term Surface Temperature (LoST) Database as a complement for GCM preindustrial simulations" By: Cuesta-Valero, et al. General comments: The development and improvement of climate models (GSM) is a leading scientific method for

understanding the Earth's climate system and its forecast- ing. Despite considerable efforts in the development of these models, there remains a large uncertainty in GCM scenarios. Authors suggest a new Long-term Surface Temperature (LoST) Database as "a reference to narrow down the spread of surface temperature climatologies on GCM preindustrial control and past millennium simulations". Preindustrial (1300-1700) ground surface temperatures cover North America and obtained from borehole temperature profiles (BTP) analysis. A robust paleotemperature evaluation technique based on extrapolating the temperature profile from the interval 200-300 m to the Earth surface has been used. Unlike BTP inversion methods, this simple technique is not burdened by the uncertainties associated with the choice of the algorithm for the inverse problem solving, and provides comparable estimates of paleotemperatures. I suppose the paper describes a new and important result for the development of climatology and can be published in the CP.

Specific comments

1. The simplicity of the technique used does not obviate the need to justify it. Strictly speaking, the extrapolation of temperature profiles from the interval of 200-300 m provides a very approximate estimate of the mean ground surface temperature in 1300-1700. It is necessary to provide a justification or refer to the paper where it was done (for example, "First-order estimate of the GST history" technique by Pickler et al., 2016).

**Although the original manuscript already cited the work of Pickler et al., we have added another reference to this paper in the description of the T0 estimates as suggested by the reviewer.**

First-order estimate technique is based on the use of formula (2), but its description ". . .the recorded temperature at a depth z can be related to an estimate of time (t)" is incorrect. Correctly: t is the time after which the temperature anomaly dT appeared at the surface reach 0.16dT at a depth of z and 0.005 at a depth of 2z. Therefore, if we assume that 0.16 dT is a negligible part of the anomaly, we should replace the

description on Figure 1

from: $z = 300m -> t \approx 1300, z = 200m -> t \approx 1700$

to: $z = 300m \rightarrow t > 700\,years\,ago, z = 200m \rightarrow t > 300\,years\,ago$

**We have made the suggested changes in Figure 1 and we have modified the conflicting definition of t in the new version of the manuscript.**

2. A large sample of BTP data was used. Obviously, many temperature profiles revealed evidences of non-climatic influences within the studied interval (hydrogeology, heterogeneity of thermal properties). Did the authors select (or correct) the initial data and by what criteria?

**Yes, the BTP database was previously filtered to remove profiles containing non-climatic signals as described in Jaume-Santero et al. (2016).**

**We have acknowledged this in the description of the borehole data.**

3. P2,L32-34: ". . .BTP measurements have been employed to estimate . . . surface flux histories over the last centuries (e.g., Beltrami, 2002; Beltrami et al., 2002, 2006)". Here the authors refer only to themselves. Meanwhile, the possibility of estimating the surface heat flux changes from ground surface temperature changes was formulated by Wang, and Bras (1999). With regard to borehole temperature data, this technique (besides the mentioned papers) was developed in (Huang, 2006; Demezhko and Gornostaeva.2015a,b).

**We have expanded our references as indicated by the reviewer, excluding the Huang (2004) work which does not employ BTP measurements for his flux estimates.**

In the last two papers an alternative measure of the Earth's climatic sensitivity has been proposed as the ratio between the ground surface flux changes and external fluxes changes. I believe that estimates of preindustrial surface heat flux changes can

also be useful for GCM simulations, as well as estimates of paleotemperatures. I would like the authors to raise this question in the "Discussion" section.

**The reviewer rises an interesting idea here; the possible estimation of the climate sensitivity using reconstructions of past changes in surface heat fluxes from BTP measurements. Such method for estimating the climate sensitivity should be further investigated, but we think that such investigation is beyond of the scope of this work.**

References Pickler, C., Beltrami, H., and Mareschal, J.-C.: Laurentide Ice Sheet basal temperatures during the last glacial cycle as inferred from borehole data, Climate of the Past, 12, 115–127, 2016. Wang, J. and Bras, R. L.: Ground heat flux estimated from surface soil temperature, J. Hydrol., 216, 214–226, 1999. Huang, S.: 1851–2004 annual heat budget of the continental landmasses, Geophys. Res. Lett., 33, L04707, doi:10.1029/2005GL025300, 2006 Demezhko, D. Y., & Gornostaeva, A. A. Late Pleistocene–Holocene ground surface heat flux changes reconstructed from borehole temperature data (the Urals, Russia). Climate of the Past, 11(4), 647-652, 2015a. Demezhko, D. Y., & Gornostaeva, A. A. Reconstructions of ground surface heat flux variations in the urals from geothermal and meteorological data. Izvestiya, Atmospheric and Oceanic Physics, 51(7), 723-736, 2015b.

**References**

**Jaume-Santero, F., Pickler, C., Beltrami, H., and Mareschal, J.-C.: North American regional climate reconstruction from ground surface temperature histories, Climate of the Past, 12, 2181-2194, 2016.**

---

## Author Response (AR1)

**Author's Response Document for "Long-term Surface Temperature (LoST) Database as a complement for GCM preindustrial simulations" by Francisco José Cuesta-Valero, Almudena García-García, Hugo Beltrami, Eduardo Zorita and Fernando Jaume-Santero.**

**We thank Dmitry Demezhko, the anonymous reviewer and Irina Rogozhina for their thoughtful and constructive feedback.**

**This Author's Response file provides a complete documentation of the changes that have been made in response to each individual Reviewer comment. Reviewer comments are shown in plain text. Author responses are shown in bold text. Corrections within the revised manuscript are shown in blue text. All page and line numbers in the author responses refer to locations in the revised manuscript.**

**The response to the Editor's comments is also included in this document.**

Anonymous Referee #1

Review of "Long-term Surface Temperature (LoST) Database as a complement for GCM preindustrial simulations" By: Cuesta-Valero, et al. Corresponding author: Beltrami

This manuscript addresses an important problem – large variation, and therefore uncertainty, in climate sensitivity estimates, that is the change in surface temperature accompanying a doubling of the atmospheric concentration of CO2. Since the 1970s the variation in estimates of ECS has not improved much; still from 1.5 to 4.5 °C.

The paper introduction provides a useful summary of why the existing GCMs have not provided convergence to a narrow band of ECS: differences in model parameterization, in particular radiative transfer and model tuning; feedback mechanisms such as ice-albedo and water vapor effects; and permafrost stability and permafrost carbon feedback. The authors suggest "a constrained preindustrial control simulation may improve the representation of those feedbacks in transient climate experiments, reducing the uncertainty of ECS estimates from model simulations, as well as reducing the spread in projections of future climate change."

The proposed constraint for such a preindustrial simulation is a database they have assembled called LoST (LoST = Long Term Surface Temperature). The paleo surface temperature at a given site is estimated by extrapolating a subsurface temperature profile from a depth range of 200 to 300 m, most sensitive to surface temperatures between about 1300 and 1700 CE, to the surface. The database is based on 514 temperature-depth profiles in North America. Creation of a LoST database offers the possibility of a better estimate of a preindustrial reference temperature field and thus an improvement in the ECS estimate.

The database is compared with five past millennium and five preindustrial control simulations from the PMIP3/CMIP5 archive to assess the realism of the simulated preindustrial

equilibrium state by the current generation of global climate models. The paper is consistent with many previous papers in advocating that borehole temperatures are a robust complement to observational (met data) and model studies of past climate.

I recommend publishing the paper after some minor to modest revisions.

Details.

1. Appropriate databases are used. With respect to borehole temperatures, it is safe to neglect heat production (does not introduce appreciable curvature into the temperature-depth profile at the depths considered) but rock heterogeneity can be more of a problem. Extrapolation of a temperature gradient from 200 m to surface to arrive at To has an error that should be discussed.

**Rock heterogeneity affects the T0 estimates via variations in thermal conductivity. In the case of inhomogeneous thermal properties, the ground temperature at a depth z (T(z)) is described as**

$$T(z) = T_0 + q_0 \cdot R(z) + T_t(t),$$  Eq. (R1)

**where $T_t$ is the surface transient perturbation, T0 is the long-term surface temperature, $q_0$ is the quasi-equilibrium heat flux and $R(z)$ the thermal resistance (Bullard, 1939). The thermal resistance requires measurements of thermal conductivity through the profile to be estimated. Unfortunately, the majority of borehole temperature profiles (BTPs) do not have conductivity measurements, which hampers the quantification of errors in T0 estimates arising from variations in thermal conductivity. Additionally, large variations of thermal conductivities can be inferred from the lithological log for each site. Generally, when large variations of thermal properties are indicated from the lithological log descriptions, rock samples are obtained, and if this is not possible, then the temperature log is not used in climate analysis. The BTP database employed in this work has been screened by Jaume-Santero et al. (2016), and only profiles suitable for climate studies were retained for the analysis.**

**As an example of a borehole site with both temperature and conductivity measurements, we provide here the T0 estimates using data from the Neil well (Canadian Arctic, see Beltrami and Taylor, 1995 for a full description of the data and the site). We find a T0 estimate of $-11.6 \pm 0.4 \,^o C$ assuming a constant thermal conductivity for the linear regression analysis for the depth range from 200 m to 300 m. If we introduce corrections by computing the thermal resistance from the thermal conductivity as a function of depth for the same depth range, we obtain a T0 estimate of $-11.4 \pm 0.2 \,^o C$. The error due to rock heterogeneity, therefore, is not large for the Neil well site. This result cannot be directly extrapolated to the rest of BTPs, since thermal properties vary from site to site, but it contextualizes the magnitude of the errors in T0 estimates in a typical borehole site due to variations of thermal conductivity with depth.**

**The new version of the manuscript describes the role of thermal conductivity in the determination of T0 values and the scarcity of thermal conductivity measurements (page 4, lines 18-25; page 8, lines 13-19).**

2. Temperatures in the depth range 200 to 300 m are largely affected by surface temperatures from 300 to 700 years prior to the temperature logging as the manuscript points out, corresponding to surface temperatures from about 1300 to 1700 CE. I would like to see a comment on how much of the signal in that depth range comes from surface temperatures outside of that time window.

**As the reviewer points out, the depth range of BTPs is fundamental to provide temporal context for the reconstructed surface temperature histories and T0 temperatures. We identify the depth range of 200-300 m with the temporal period 1300-1700 of the Common Era, but the effect of the Little Ice Age and the Medieval Warm Period may also affect the T0 estimates. However, the spatial extent of both events is not homogeneous over North America, which implies that not all BTPs employed here are affected by these events. Additionally, their influence should be part of any transient millennial-scale climate simulation and thus, these climate events should be represented within both the transient simulations and the BTPs.**

**We have added a few lines commenting on the impact of the Little Ice Age and the Medieval Warm Period on the T0 estimates (page 8, lines 19-25).**

3. Figure 1 is a good illustration of the extrapolation of the borehole temperature profile to the surface. The term in the caption "linear fit of the last 100 m" is ambiguous. Say, "linear fit of bottom 100 m" or better still, "linear fit of temperatures between 200 and 300 m." The paper should also say that any thermal conductivity heterogeneity in the depth range 0 to 200 m would affect the zero depth (i.e. Surface) extrapolated temperature and ideally give a bound for how big an error that would introduce.

**We have changed the caption of Figure 1 in the new version of the manuscript. Regarding the effect of the thermal conductivity heterogeneity on T0 estimates, please see our answer to the first comment.**

4. Figure 2. Is the temperature scale on Fig 2(b) mislabeled or is it some kind of a non-linear scale? The colored temperature scale for Fig 2(c) and (d) needs a label and units.

**Indeed, Fig. 2(b) was mislabeled in the previous version of the manuscript, and the temperature scale of Fig. 2(c, d) should read "Surface Temperature (ºC)". We have corrected all these issues in the new version of the figure.**

5. The paper would be improved by a discussion of various kinds of uncertainties in LoST and whether the magnitude of those uncertainties detract significantly from the goal of providing a robust preindustrial surface temperature field. Include: (a) extrapolation uncertainties for a typical borehole site. (b) whether the 514 sites are generally representative of the topography (elevation and site azimuth) of the region being modeled (scatter in extrapolated borehole temperatures in a region can vary by ~ 4 oC). (c) Are the BTT's in Fig 2(b) corrected for elevation or are elevation differences at particular latitude (considerable in North America) the cause of about 10 oC scatter at constant elevation?

**We have included a paragraph in the Discussion section addressing the reviewer's points (from line 12 in page 8 to line 2 in page 9).**

Overall this is a refreshing new approach of showing how borehole temperature profiles can be used to complement the more conventional meteorological and GCM modeling studies to reveal the long-term evolution of surface temperature on the planet.

Dmitry Demezhko (Referee)

Review of "Long-term Surface Temperature (LoST) Database as a complement for GCM preindustrial simulations" By: Cuesta-Valero, et al.

General comments: The development and improvement of climate models (GSM) is a leading scientific method for understanding the Earth's climate system and its forecast- ing. Despite considerable efforts in the development of these models, there remains a large uncertainty in GCM scenarios. Authors suggest a new Long-term Surface Temperature (LoST) Database as "a reference to narrow down the spread of surface temperature climatologies on GCM preindustrial control and past millennium simulations". Preindustrial (1300-1700) ground surface temperatures cover North America and obtained from borehole temperature profiles (BTP) analysis. A robust paleotemperature evaluation technique based on extrapolating the temperature profile from the interval 200-300 m to the Earth surface has been used. Unlike BTP inversion methods, this simple technique is not burdened by the uncertainties associated with the choice of the algorithm for the inverse problem solving, and provides comparable estimates of paleotemperatures. I suppose the paper describes a new and important result for the development of climatology and can be published in the CP.

Specific comments

1. The simplicity of the technique used does not obviate the need to justify it. Strictly speaking, the extrapolation of temperature profiles from the interval of 200-300 m provides a very approximate estimate of the mean ground surface temperature in 1300-1700. It is necessary to provide a justification or refer to the paper where it was done (for example, "First-order estimate of the GST history" technique by Pickler et al., 2016).

**Although the original manuscript already cited the work of Pickler et al., we have added another reference to this paper in the description of the T0 estimates as suggested by the reviewer (page 5, line 2).**

First-order estimate technique is based on the use of formula (2), but its description ". . .the recorded temperature at a depth z can be related to an estimate of time (t)" is incorrect. Correctly: t is the time after which the temperature anomaly dT appeared at the surface reach 0.16dT at a depth of z and 0.005 at a depth of 2z. Therefore, if we assume that 0.16 dT is a negligible part of the anomaly, we should replace the description on Figure 1

from: $z=300m$->$t\approx1300, z=200m$->$t\approx1700$

to: $z=300m \rightarrow t>700 years ago, z=200m \rightarrow t>300 years ago$

**We have made the suggested changes in Figure 1 and we have modified the conflicting definition of t in the new version of the manuscript (page 5, lines 4-5).**

2. A large sample of BTP data was used. Obviously, many temperature profiles revealed evidences of non-climatic influences within the studied interval (hydrogeology, heterogeneity of thermal properties). Did the authors select (or correct) the initial data and by what criteria?

**Yes, the BTP database was previously filtered to remove profiles containing non-climatic signals as described in Jaume-Santero et al. (2016).**

**We have acknowledged this in the description of the borehole data (page 4, lines 9-12).**

3. P2,L32-34: ". . .BTP measurements have been employed to estimate . . . surface flux histories over the last centuries (e.g., Beltrami, 2002; Beltrami et al., 2002, 2006)". Here the authors refer only to themselves. Meanwhile, the possibility of estimating the surface heat flux changes from ground surface temperature changes was formulated by Wang, and Bras (1999).

With regard to borehole temperature data, this technique (besides the mentioned papers) was developed in (Huang, 2006; Demezhko and Gornostaeva.2015a,b).

**We have expanded our references as indicated by the reviewer, excluding the Huang (2004) work which does not employ BTP measurements for his flux estimates (page 2, lines 33-35).**

In the last two papers an alternative measure of the Earth's climatic sensitivity has been proposed as the ratio between the ground surface flux changes and external fluxes changes. I believe that estimates of preindustrial surface heat flux changes can also be useful for GCM simulations, as well as estimates of paleotemperatures. I would like the authors to raise this question in the "Discussion" section.

**The reviewer rises an interesting idea here; the possible estimation of the climate sensitivity using reconstructions of past changes in surface heat fluxes from BTP measurements. Such method for estimating the climate sensitivity should be further investigated, but we think that such investigation is beyond of the scope of this work.**

**As suggested by the Editor, we have included a paragraph in the Discussion section commenting on the differences between proxy and borehole techniques and the role of the LoST database as a complementary tool for evaluating climate simulations of preindustrial times (from line 33 in page 9 to line 10 in page 10).**

**References**

[revised manuscript text omitted]

**Two Sigma Values**

0.0   0.1   0.2   0.3   0.4   0.5   0.6   0.7   0.8   0.9 (ºC)

**Figure S6.** Errors ($2\sigma$ values) of LoST temperatures estimated as described in Section S1. The spatial average is 0.2 °C.

[Figure]

**Figure S7.** Trends of the difference between air and ground (1.0 m depth) temperatures from PMIP3/CMIP5 simulations. (a) Results for PM simulations (1300-1700 CE). (b) Results for piControl simulations. **9**

[Figure]

**Figure S8.** Altitude distribution over the LoST domain (black histogram) and at grid cells containing BTPs (red histogram) from the ETOPO2 product.

---

## Author Response (AR2)

**Response to Editor's Document for "Long-term Surface Temperature (LoST) Database as a complement for GCM preindustrial simulations" by Francisco José Cuesta-Valero, Almudena García-García, Hugo Beltrami, Eduardo Zorita and Fernando Jaume-Santero.**

**We thank Irina Rogozhina for her constructive feedback.**

**This file provides a complete documentation of the changes that have been made in response to the editor's comments. Editor comments are shown in plain text. Author responses are shown in bold text. Corrections within the revised manuscript are shown in blue text. All page and line numbers in the author responses refer to locations in the revised manuscript.**

Editor Decision: Publish subject to minor revisions (review by editor) (10 May 2019) by Irina Rogozhina

Comments to the Author:

Dear authors,

From my point of view, you have addressed most of the reviewers' comments in a satisfactory way but there are still some issues that should be considered prior to the final publication. In the following I am referring to the page and line numbers in the reviewed manuscript with highlighted changes.

Page 5, lines 15-16: This sentence is somewhat misleading. I assume that you mean: Influences of long-term perturbations of the past surface energy budget outside of that temporal window may also affect the temperature in the employed depth range. Your current phrasing may suggest that you present methods for detection of such signals or at least some evidence of detected signals. Since this is not the case, I advise to rephrase this sentence.

**We have change that in the new version of the manuscript (see page 5, lines 12-13).**

Page 8, lines 17-20: It may be impractical indeed, but one needs to have at least an approximate estimate of the errors. There must be more measurements than from just one borehole that you examine in your response or some reference to add. If there is no such reference, I would expect that you would use the entire subset of existing measurements of thermal conductivity profiles to provide an estimate of potential errors, since your error estimate from a single measurement may be biased (by your choice of homogeneous values or insufficient heterogeneity in the thermal properties of the selected borehole). Hence, I suggest that you either include at least some estimates in the supplementary materials or outline why such analysis is technically too demanding.

**Although such an analysis would be very interesting, the database employed in this article (Jaume-Santero et al. 2016) does not include any measurements of thermal conductivity. Additionally, vertical variations of thermal conductivity tend to be distributed around an average value, as it can be seen in the example provided for the Response to Reviewers (Neil Well, Beltrami and Taylor 1995). It is also worth to mention that temperature logs are routinely screened before performing any climate analysis with them. If the thermal conductivity varies systematically with depth (e.g., linearly), this will induce unexpected temperature variations in the logs, making them unsuitable for climate studies. Those logs are removed from the database during the screening process (see Jaume-Santero et al. 2016 for details) and thus the logs with spurious signals due to changes in thermal conductivity are not included in this analysis. Therefore, it is reasonable to consider a constant value for thermal conductivity.**

**We have included this explanation in the new version of the manuscript (page 8, lines 14-19).**

Page 8, line 21-23: Although this sentence is certainly true, it calls for a supporting reference and a clarification of the amplitude of heterogeneity at play. In particular, to which extent may it affect your estimates when they are compared to the Picontrol?

**Indeed, piControl simulations do not include a representation of the little ice age or the medieval warm period, which is probably contributing to the slightly poorer agreement between piControl temperatures and LoST data in comparison with the past millennium case (Figure 3). Nonetheless, the temperature climatologies from both piControl and past millennium simulations are comparable to our LoST temperatures.**

**We have added this explanation and the requested references on page 8, lines 23-24 and 27-29.**

Finally, I spotted some typos/missing info throughout the manuscript (some examples below). Please, make sure that you have carefully copy-edited the manuscript before publication.

Page 2, line 28: "an useful" => "a useful"

Page 3, line 1: unite Gonzalez-Rouco et al., 2006, 2009.

Page 3, line 18: Add the link

Page 3, line 27: "surface temperature" => "surface temperatures"

Page 8, line 22: "extend" => "extent"

Page 9, line 27: Papers in preparation should not be cited.

Page 10, line 7: "captures" => "capture"

**We have corrected all these typos and provided with a link to the database.**

Please, contact me if you need any further clarifications. I look forward to receiving a revised version of your manuscript.

Kind regards,

Irina

**References**

[revised manuscript text omitted]